# 3D Positioning of Drones through Images

**DOI:** 10.3390/s24175491

**Published:** 2024-08-24

**Authors:** Jianxing Yang, Enhui Zheng, Jiqi Fan, Yuwen Yao

**Affiliations:** School of Mechanical and Electrical Engineering, China Jiliang University, Hangzhou 310018, China; s22010811041@cjlu.edu.cn (J.Y.); p22010854026@cjlu.edu.cn (J.F.); p22010854149@cjlu.edu.cn (Y.Y.)

**Keywords:** unmanned aerial vehicle (UAV), deep learning, drone height calculation, positioning

## Abstract

Drones traditionally rely on satellite signals for positioning and altitude. However, when in a special denial environment, satellite communication is interrupted, and the traditional positioning and height determination methods face challenges. We made a dataset at the height of 80–200 m and proposed a multi-scale input network. The positioning index RDS achieved 76.3 points, and the positioning accuracy within 20 m was 81.7%. This paper proposes a method to judge the height by image alone, without the support of other sensor data. One height judgment can be made per single image. Based on the UAV image–satellite image matching positioning technology, by calculating the actual area represented by the UAV image in real space, combined with the fixed parameters of the optical camera, the actual height of the UAV flight is calculated, which is 80–200 m, and the relative error rate of height is 18.1%.

## 1. Introduction

Unmanned Aerial Vehicles (UAVs) have broad application prospects in numerous fields, including surveying and mapping [1], agriculture [2], rescue operations [3], and military applications. However, during actual flight operations, UAVs often encounter “denial environments” where GPS signals are lost or interfered with, resulting in the loss of precise positioning and altitude determination capabilities. This not only greatly reduces the operational efficiency and reliability of UAVs but may also lead to serious consequences such as loss of control. Therefore, achieving autonomous positioning and assisted altitude determination for UAVs in denial environments has long been a major challenge in this field.

Cross-View Geo-Localization based on computer vision technology is one effective approach to address this issue. Initially, the University-1652 dataset [4] introduced the use of drone views for cross-view geo-localization. University-1652 proposed tasks such as drone view target localization and drone navigation, expanding the application scope of drone geo-localization. The basic idea is to estimate the precise location of the UAV by matching the drone’s overhead imagery with satellite orthoimages of known locations. Early image retrieval localization methods used metric learning to continuously bring the feature vectors of similar areas in drone and satellite images closer together while pushing apart the feature vectors of different areas, thus achieving image matching. This method requires preparing a vast amount of satellite imagery in advance, segmenting each image, extracting features, encoding features, and storing them, resulting in enormous computational and storage costs. Moreover, in practical applications, drone images often cannot be perfectly aligned with the images in the database, leading to localization errors.

To address these issues, the authors proposed the FPI (Finding Point with Image) [5], a novel end-to-end localization paradigm. FPI borrows ideas from the single object tracking domain, using both drone and satellite images as input. It extracts features from both images through a network, calculates the similarity between their features, and generates a heatmap. The drone’s position corresponds to the peak point in the heatmap. By utilizing a single backbone to simultaneously process drone and satellite images, it integrates traditional feature extraction and relationship modeling. During the feature extraction process, it effectively utilizes the powerful performance of the Transformer mechanism, enabling information interaction between drone and satellite features. The characteristics of this method are: (1) In the early stages of feature extraction, the model can determine which relevant features to retain, reducing the loss of target information. (2) The Transformer mechanism establishes interaction channels between drone and satellite images, thereby improving performance. (3) Feature maps of different sizes are generated and preserved at each stage, making subsequent interaction between drone and satellite features possible.

While the above methods have achieved good results, several problems still exist: (1) The network training process is based on drone images captured at 80–100 m altitude. At this height, accuracy is relatively high, but when the altitude changes, the features of same-sized drone images change, and the accuracy of models trained on 80–100 m altitude images decreases significantly, indicating poor model generalization. (2) During UAV flight, satellite signals assist in determining flight altitude. In denial environments, the z-direction GPS signal is also lost, necessitating a method to assist in determining the flight altitude.

How do UAVs determine their altitude during flight? The current mainstream approach is multi-sensor data fusion, including accelerometers and gyroscopes, barometers, GNSS, and ultrasonic rangefinders. Multi-sensor fusion methods for altitude determination have achieved good accuracy, but each individual sensor has its own limitations.

Accelerometers and gyroscopes are low-cost and widely carried by UAVs, but the altitude they measure is obtained through double integration calculations, which leads to continuously accumulating errors that increase over time. Additionally, the vibrations of the UAV during movement affect the operation of accelerometers and gyroscopes.

Barometers use pressure sensors to measure the altitude of the UAV during flight. Their advantages include relatively low cost and good real-time performance, providing immediate altitude data without complex data processing. However, barometers are affected by weather conditions (such as pressure changes, temperature variations, etc.). In conditions of significant pressure changes and extreme weather, the accuracy of barometers decreases. Moreover, barometers cannot provide absolute altitude. They only provide relative altitude information, i.e., height relative to ground atmospheric pressure. Therefore, in most application scenarios, they need to be combined with other altitude measurement techniques.

Compared to accelerometers and barometers, GPS obtains altitude information through distance intersections without accumulating errors. However, GPS performs poorly in z-direction altitude measurement compared to its horizontal positioning accuracy. In the denial environment proposed in this paper, GPS signals face risks of interruption and loss.

In this paper, we expanded the range of the dataset, with drone image capture heights ranging from 80 m to 200 m. Specific details will be introduced in Section 2.3. Considering that drone images at different altitudes have features of different natures, based on OS-FPI(An overall architecture for extracting UAV and satellite image features in a single channel. In Section 2.2, we will provide further introduction) [6], we designed a dual-stream network with multi-scale drone image input. After obtaining a drone image, we crop input 1 and input 2 of sizes 256 × 256 and 152 × 152, respectively, from the center point of the drone image, and interact them with the satellite image separately. The significance of this approach is that the smaller-sized satellite image represents low-altitude drone images with richer details, while the larger-sized satellite image represents high-altitude drone images with broader ground features. Multi-scale input has a good effect on solving localization when the drone altitude varies between 80 m and 200 m.

In addition, we propose a method for assisting altitude estimation through image analysis. In ground images captured vertically downward by a UAV, if the actual ground distance between two points at a considerable distance in the image is known, the flight altitude can be calculated using the fixed parameters of the camera. Therefore, we crop a drone image into left and right halves, input them separately into the network, and calculate the latitude and longitude information of the center points of the left and right images, thereby obtaining the absolute ground distance represented by the original drone image. Finally, combined with camera parameters, we estimate the absolute altitude of the UAV above the ground.

The contributions of this paper are summarized as follows:

1. Based on the UL14 dataset [6], we added more drone images at various altitudes, expanding from 80 to 100 m to 80 to 200 m. We designed a dual-stream network with multi-scale drone image input, inputting two drone images with the same center point and using a cross-attention mechanism for the two inputs. This improved the model’s generalization within the 80m–200m range, achieving the RDS metric of 76.3.

2. We proposed an image-assisted altitude determination method to compensate for the loss of z-direction GPS signals in denial environments. Within the 80–200 m range, the total error rate is 18.1%.

## 2. Related Work

### 2.1. Overview of UAV Geolocation Methods

Cross-view geolocation technology primarily refers to matching and comparing images from different perspectives. In this paper, it specifically refers to matching UAV images with satellite images to locate the position of the UAV image on the satellite image, thereby geolocating the UAV. Traditional cross-view image matching methods, such as SIFT, SURF, and ORB, often struggle to meet requirements due to differences in image sources.

However, the rapid development of neural networks and their application in image matching has made cross-view geolocation technology feasible for practical positioning. Compared to traditional ground images, satellite and remote sensing images cover larger areas. In early work, Lin [7,8,9] matched street-view images with aerial images to obtain location information. Tian et al. [10] introduced neural networks for matching and adopted a Faster R-CNN structure.

These methods achieved some success, but due to significant feature differences between images from different sources, extracting and matching features from these diverse sources remains a primary challenge in this field. Y. Shi [11] proposed aligning aerial images with satellite images to reduce differences, mainly using polar coordinate transformation.

To reduce matching errors caused by perspective changes, A. Toker [12] used generative adversarial networks to convert street-view images into drone images. Y. Zhu [13] introduced GeoNet, which models by fully utilizing spatial hierarchies to learn common features in images from different perspectives. These methods employ traditional approaches to narrow the differences between images from various viewpoints, thereby achieving cross-view data matching.

### 2.2. Transformer

Currently, deep learning is thriving in image processing, with applications in image classification, object detection, semantic segmentation, and more. Within deep learning, there is a distinction between traditional convolutional neural networks and transformer-based architectures. Traditional CNNs, such as CNN [14], AlexNet [15] and VGG [16], use convolutional layers, pooling layers, and fully connected layers to extract features from images and perform classification or regression tasks. As transformers developed in natural language processing, researchers introduced them to image processing.

The ViT model, proposed by Dosovitskiy et al. [17] in 2020 for image classification, divides images into a series of patches, serializes these patches, and inputs them into a Transformer for processing, with final classification through fully connected layers. The DeiT model, proposed by Touvron et al. [18] in 2020 for image classification, achieves efficient classification on small-scale datasets through pre-training on large-scale image datasets and knowledge distillation to improve generalization.

Wang et al. [19] proposed the PVT structure in 2021, combining traditional pyramid features with Transformer advantages to capture feature information at different scales and levels. The PVT model introduces multiple parallel pyramid modules, each composed of multiple levels with a Transformer module at each level. These levels have different resolutions and receptive fields, capturing feature information at various scales through self-attention mechanisms. PCPVT, an improved version of PVT for image classification tasks, incorporates cross-layer connections in each level’s context feature fusion module to merge context features from previous levels with current level features. This cross-layer connection design gradually introduces higher-level context information and improves feature representation through feature fusion.

Image retrieval methods for cross-view geolocation [20] require strict alignment between query images and database images, which is challenging in specific situations. Dai [21] applied this structure to the field of drone image matching and subsequently Chen [6] proposed an FPI method, a new end-to-end approach for direct point searching, using a dual-stream network without shared weights for UAV and satellite image views. A response map is then generated through relationship modeling, with the point of maximum response being the model’s predicted current location of the UAV image.

In OS-FPI [6], the authors proposed a single-stream network. Feature extraction for both UAV and satellite images uses the same backbone. At different stages of feature extraction, the interactivity of Transformers is utilized to establish information exchange between UAV and satellite features. Its characteristics include: (1) Using the Transformer mechanism to establish connections between UAV and satellite images, enabling information exchange and improving accuracy. (2) Retaining feature maps of different sizes at each stage to establish more interactions and connections between subsequent features.

### 2.3. UL14plus Dataset

Currently, commonly used datasets for cross-view matching geolocation include the CVUSA dataset, CVACT dataset, and University1652 dataset. Although there are many mature cross-view geolocation datasets available for geolocation tasks, there are few relevant datasets that can be applied to visual localization on UAV platforms. While the recent University1652 pioneered the inclusion of UAV perspective images in cross-view geolocation datasets, its inherent limitations prevent its application to pure visual geolocation tasks for UAVs.

Firstly, the images in the University1652 dataset are collected from buildings at universities around the world, making them spatially discrete (Figure 1). In contrast, the ground images captured by onboard cameras for UAV autonomous positioning in GNSS environments, as proposed in this paper, are spatially continuous. Secondly, the majority of the content in this dataset’s images consists of buildings, while actual UAV visual localization scenes include many elements such as roads and vegetation in addition to buildings. Finally, to save labor costs in the creation of the University1652 dataset, the UAV perspective images were collected from 3D model buildings in Google Earth. Although the modeling detail in Google Earth is high, there are still significant differences compared to images captured by optical cameras in real-world scenarios.

Therefore, although the University1652 dataset introduced UAV platform perspective images and proposed a UAV perspective target localization task, it cannot serve as a dataset for the UAV cross-view matching geolocation task described in this paper. To address the deficiencies in existing geolocation datasets and to cater to the characteristics of the UAV cross-view matching geolocation algorithm proposed in this paper, we introduce the UL14plus (University Localization14plus) dataset. Based on the DenseUAV and UL14 datasets, it aims to support UAV self-localization and altitude determination. Previous datasets like DenseUAV and UL14 focused more on latitude and longitude information for positioning, with less emphasis on altitude data.

The UL14 dataset is a reconstruction based on the DenseUAV dataset. Unlike DenseUAV’s image retrieval task, UL14 is tailored for the FPI benchmark task. It employs a data augmentation method that allows search images to have both random scale and random target distribution characteristics. Two variables were introduced to crop satellite images: a center point coverage C (initial value 0.75) and a scale range (512–1000). C determines the spacing of UAV images within the satellite image. A smaller C concentrates UAV images in the central region, while a larger C disperses them. The scale parameter determines the satellite image’s scale in actual space. A larger scale means each pixel represents a greater actual spatial distance. This data augmentation method aims to enhance the model’s resistance to scale changes and offsets.

UL14plus is developed based on UL14, specifically for the multi-altitude localization/altitude determination benchmark proposed in this paper.

UAV images were captured using a DJI Phantom 4 Pro drone, flying at altitudes of 120–200 m above ground (in 20-m intervals), as shown on the right side of Figure 2. The DJI Pilot program’s waypoint flight feature was used for photography, ensuring shots were taken at the same waypoint positions across different altitudes. Choosing university areas for shooting offers two advantages: 1. The buildings and other landscape features in universities are distinct, facilitating subsequent feature extraction by deep learning models; 2. Buildings on campuses are generally not too tall, ensuring the safety of drone flight paths during image capture. To ensure spatial continuity and dense sampling in the dataset construction, the drone’s flight path design considered sampling density, with a sampling distance of 20 m, contributing to the uniform distribution of the dataset. Additionally, overlapping portions in images from adjacent shooting points help the model enhance its ability to discern subtle differences.

As shown on the left side of Figure 2, three groups of images represent scenes captured at the T − 1, T, and T + 1 waypoints. Each image overlaps with adjacent images. In previous image retrieval methods, these three images might have been localized to the same position. However, in the point-by-image localization algorithm proposed in this paper, the centers of these three images should be localized to three different geographical locations. Therefore, using a dense sampling method to create the dataset helps the subsequently designed cross-view matching geolocation algorithm learn the offset differences between images, enabling the model to develop a strong spatial position information perception ability. This contributes to improving the model’s localization accuracy and forms the basis for the altitude determination method proposed in this paper.

During data collection, the UAV’s gimbal was pointed vertically downward, capturing vertical overhead views at each waypoint. For different altitudes at the same waypoint, horizontal offset was controlled within 1 m, and vertical offset within 0.5 m. The gimbal camera had an FOV of 84° and a focal length of 8.8 mm, with the zoom set to minimum during shooting. Photo dimensions were 4864 × 3648 pixels.

Training and test data were collected from four universities in Zhejiang, China.

The training set at 120–200 m altitude includes 712 sampling points, 3560 UAV images, and 3560 satellite images. At 80–100 m altitude, a portion of the UL14 dataset was selected, including 334 sampling points, 1002 UAV images, and 1002 satellite images. In the training set, the ratio of satellite to UAV images is 1:1. The UL14plus dataset still includes some content from the UL14 dataset, which is significant because: First, it includes UAV perspective images from multiple altitudes, thus training a model more adaptable to altitude changes; Second, the new and old UAV images differ noticeably in capture time, style, and weather conditions, helping to enhance the model’s generalization ability.

UAV images were captured in real scenes at various flight altitudes. The resolution of UAV images is (4864, 3648). To improve the model’s adaptability to multiple altitudes, we crop UAV images from different heights with different parameters to simulate images from more altitudes. We aim to build a dataset encompassing ground features under more conditions. Figure 3 below shows the cropping situation for UAV images at 200 m altitude, where c is the cropping ratio. All cropped images share a common center point. The crop size gradually decreases. In subsequent model calculations, these UAV images will be reshaped to a uniform size of 256 × 256. Therefore, the larger cropped images will contain a wider range of ground features, while smaller cropped images will contain more ground details. This improves the model’s adaptability to multi-scale features. When UAV images are from 80 to 100 m, no cropping is conducted to ensure sufficient large-scale ground features in the images.

Satellite views use Google Maps’ level 20 images. Based on the actual latitude and longitude of the UAV shots, the UAV shooting point is located on a satellite image covering the entire school. Centered on this point, a (4000, 4000) satellite image is cropped and reshaped to a resolution of (512, 512).

The test set at 120–200 m altitude includes 120 sampling points, 600 UAV images, and 7200 satellite images. The test set at 80–100 m altitude includes part of the UL14 dataset’s test set, with 100 sampling points, 300 UAV images, and 3600 satellite images. In the test set, each UAV image corresponds to 12 satellite images containing that UAV image. These 12 satellite images are local crops from a satellite image covering the entire school. The UAV image is positioned differently on each of the 12 satellite images. The satellite image crop sizes range from 700 to 1800, corresponding to actual coverage areas with side lengths between 180 m and 463 m. The testing phase involves locating the UAV image on the 12 satellite images. In creating the test set, we considered real-world challenges in cross-view matching geolocation such as scale transformations and position offsets. Therefore, each UAV aerial photo corresponds to 12 satellite images of different spatial scales, and the position of the UAV image within the satellite image is randomly generated (Figure 4). This test set design increases the difficulty of geolocation, better aligning with actual UAV working scenarios.

## 3. Methods

This section introduces the MH-FPI structure improved based on OSPCPVT and an image-based altitude determination method. In Part A, we introduce the overall structure of our model, MH-FPI. Part B describes the optimized attention structure. In Part C, we present a method for determining altitude using UAV images.

### 3.1. Overall Structure

Given a set of UAV images and a satellite image containing the UAV image, we aim to locate the position of the UAV image within the satellite image. Assuming the satellite image is already stored in a database with calibrated coordinates, we can calculate the UAV image’s position to achieve localization. As shown in Figure 5, the backbone network for feature extraction is designed for UAV positioning and altitude determination at multiple heights, which we call MH-PCPVT. Compared to the backbone OS-PCPVT from OS-FPI, we have added an additional dimension of input.

The input consists of one satellite image and two UAV images. The relationship between Drone and drone1 is as follows: Drone is a 256 × 256 UAV image, while drone1 is a smaller 128 × 128 UAV image cropped from the center of the Drone.

The overall architecture of this paper is shown in Figure 7. The input to the backbone network consists of two UAV images of size 96 × 96 × 3 and one satellite image of size 384 × 384 × 3. These three images are divided into 4 × 4 × 3 patches. Through linear transformation, they are reshaped into vectors of size N² × C1 and N² × C1. The patches are then merged and input into the Transformer encoder for feature extraction, establishing communication channels between UAV and satellite features for information exchange.

After the first Transformer encoder in each stage, a Position Encoding Generator (PEG) is employed. With each stage, the size of the feature maps decreases while the number of channels increases.

After the feature maps of three scales are extracted through the backbone network, these feature maps are used in the feature fusion network, and the information interaction between the UAV and the satellite features is established. The feature map of the satellite map adopts the feature pyramid structure. Feature maps of different scales have different levels of semantic information. When feature map scales are compressed, deep feature maps may be damaged, but shallow feature maps can be preserved. Therefore, the feature pyramid structure is used to recover the compressed feature map and finally fuse the information to improve the accuracy of the result. The feature maps of the three branches are fused to obtain a heat map. By observing the heat distribution results in the heat map, the feature map is finally restored to the size of the satellite map by interpolation, and the relative position of the UAV on the satellite view is obtained.

### 3.2. Feature Extraction Network

In this section, we propose a backbone network MH-PCPVT that simultaneously extracts input image features. The baseline of the model is twin-PCPVT-S, and we have made changes on this basis, mainly in the input part and the attention structure part. Our backbone structure provides a channel for information exchange between the satellite image and the two drone images at the same time as feature extraction, thus improving the efficiency of feature extraction.

As shown in Figure 6, the input of MH-FPI is two drone images with a size of 96 × 96 × 3 and a satellite image with a size of 384 × 384 × 3. We divided the drone image into 24 patches and the satellite image into 96 patches, each with a size of 4 × 4 × 3 (transformer stage in Figure 7). We then fed them into a linear projection, reshaping them into two-dimensional patches with sizes $24^{2}$ × C1 and $96^{2}$ × C1 (C1 is the number of channels in the first stage). Finally, we combined them to create an embedded patch with a size of ($24^{2}$ + $24^{2}$ + $96^{2}$) × C1.

Next, we input the merged patch into the Transformer block for feature extraction. After each Transformer block, the absolute position coding is replaced by the position-coding generator (PEG) module [22]. Conditional position coding (CPE) [22] is implemented by the position-coding generator PEG, which is more flexible for input sequences and also preserves translation invariance. This feature has a positive effect on training.

After each stage of the MH-PCPVT network, the output is reshaped into a feature map. In the first stage, the size of the drone feature map is 24 × 24 × C1, while the size of the satellite feature map is 96 × 96 × C1. The current output feature map is then used as input for subsequent stages and the process is repeated.

After three stages, each stage generates one feature map of the satellite image and two feature maps of the drone image, which are compressed 4, 8 and 16 times compared to the original image, respectively. Originally, the last layer of the network would continue to compress the feature map once, which is 32 times more compressed than the original map, which is not adopted by us. The reason is that too small a feature map is not conducive to subsequent restoration and is not conducive to the model learning the correlation between drone and satellite images. These compressed feature maps are then fused to produce a final prediction.

### 3.3. Improved Attention Structure

For the multi-altitude positioning proposed in this paper, we aim to enhance the model’s adaptability to changes in spatial height during localization. To this end, we designed the aforementioned multi-scale input of UAV images. Using UAV images with the same center point, a 256 × 256 UAV image, when reshaped to a uniform size of 96 × 96, provides a wider range of ground features. A 128 × 128 UAV image, when reshaped to 96 × 96, offers richer detail features.

How can the model compare the two UAV images to identify the UAV’s position on the satellite image? We achieve this through an improved attention structure. At each stage of the backbone network, as the model extracts features, information exchange occurs between the UAV images and the satellite image.

The design concept is to first perform attention interaction between the larger 256 × 256 UAV image and the satellite image, creating an area of focus on the satellite image. Then, the smaller 128 × 128 UAV image interacts again with the satellite image that now has a focused area, to obtain a more precise matching position.

The right side of Figure 7 shows the attention mechanism part. In transforming input patches into K and V, we continue to use the SRA module [19] method to reduce computational load. First, the drone and satellite patches are transformed into Qu, Ku, Vu for the UAV and Qs, Ks, Vs for the satellite image through linear transformation and SRA. The UAV image first undergoes self-attention, followed by cross-attention between the UAV and satellite images. The vector obtained after cross-attention is concatenated with the drone1 vector, and the process of linear transformation, self-attention, and cross-attention is repeated. Finally, the vectors obtained from the two self-attention operations and the last cross-attention operation are concatenated to form the output of the attention operation.

MH-PCPVT adopts asymmetric cross-attention during the operation of the attention mechanism. Self-attention operations are allowed for each sequence, and the combination of feature extraction and relationship modeling is realized through sequence concatenation and cross-attention operations. In the process of attention calculation, the UAV features perform self-attention. The advantages are: first, a full crossover operation will require more computation; second, the self-attention process of the UAV image enhances the information of the UAV feature branch. Through the method of joint feature extraction and information exchange, the satellite image and two UAV images establish information exchange, so that the positioning performance is improved.

### 3.4. UAV Image-Based Altitude Determination Method

When a UAV is in flight with its camera pointing vertically downward and the focal length unchanged, as shown in Figure 8, there is a relationship between the camera’s field of view and the flight altitude. By determining the actual area of a certain region in the UAV image, the actual flight altitude can be calculated.

This study used the DJI Phantom 4 Pro drone for image capture. The specific camera parameters are FOV 84°, 8.8 mm/24 mm (35 mm format equivalent), f/2.8–f/11 with autofocus. When capturing images, the camera uses the minimum focal length (minimum magnification), with a vertical downward view, and a 4:3 aspect ratio. According to official DJI data, under 4:3 aspect ratio conditions, the actual shooting area at 100 m altitude is 15,542.7 m², and at 500 m altitude, it is 388,567.9 m². From this, we can deduce that the photo shooting area has a linear relationship with the square of the flight altitude, with *S*:$H^{2}$ = $S_{1}$:$H_{1}^{2}$ = 1.55427:1.

Given this known ratio between the photo shooting area and the square of the flight altitude, we can obtain the coordinates of four points in the original image through our model, calculate the area of the square formed by these 4 points, and thus calculate the flight altitude. Figure 9 shows the altitude calculation process. After a drone captures an image, it is cropped into four smaller images. These are separately input into the model to calculate the position of each small image’s center point, thus obtaining the coordinates of the small image centers. We then calculate the area of the small square block in the drone image on the right side of Figure 9, restore the overall image area, and finally calculate the flight altitude through the ratio.

During the process of image cropping–model processing–coordinate generation, due to the cropping of the image, the perspective shifts, differing from the vertical downward view during training. The resulting predicted coordinate accuracy makes it difficult to achieve the precision of standalone testing. To address this, we will introduce the image cropping–model processing–coordinate generation process and the corresponding optimization process using Sequential Quadratic Programming (SQP).

In the dataset Section 2.3, we introduced the composition of the training and test sets. In the test set, one UAV image corresponds to 12 satellite images of different scales, all containing the UAV image but with the UAV image at different positions within the satellite images (Figure 3). Through the model, 12 latitude and longitude predictions are generated, with different localization accuracies for satellite images of different scales. In our proposed altitude calculation method, we aim to calculate the area of the drone’s shooting region as accurately as possible, and then obtain the drone’s flight altitude through a linear relationship. We transform this problem into how to optimize the solution for the area of the small square block in the drone image on the right side of Figure 9. Under the vertical perspective of the drone, the positioning accuracy can generally achieve an RDS75 score. Therefore, we consider the coordinates of the image center as accurate values, and view the predicted latitude and longitude coordinates of the cropped drone images from the top-left, bottom-left, top-right, and bottom-right corners as objects to be optimized. The optimization process uses Sequential Quadratic Programming (SQP).

1. Initialization

· Drone image center point coordinates ($x_{c}$, $y_{c}$)

· Values to be solved for the cropped drone images from top-left, bottom-left, top-right, and bottom-right corners: ($x_{1}$, $y_{1}$), ($x_{2}$, $y_{2}$), ($x_{3}$, $y_{3}$), ($x_{4}$, $y_{4}$).

· Cropped drone images *i* (*i* = 1, 2, 3, 4) from top-left, bottom-left, top-right, and bottom-right corners.

m predicted coordinates ($x_{ij}$, $y_{ij}$), *j* = 1, 2, *…*, *m*, *m*≤ 12 (after eliminating outliers)

Known accuracy probability p1 for each predicted coordinate *($p_{j}$)* takes the RDS score of prediction accuracy for satellite images of different sizes)

2. Construct the objective function:(1)$$J=\sum_{i=1}^{4}\sum_{j=1}^{m}p_{j}\sqrt{{\left(x_{i}-x_{ij}\right)}^{2}+{\left(y_{i}-y_{ij}\right)}^{2}}$$

The objective function sums the weighted deviations between multiple predicted coordinates for each vertex and the true coordinates (the unknowns to be solved).

The physical meaning of minimizing *J* is as follows: Under the constraints of all predicted coordinates, find an optimal set of coordinates ($x_{1}$, $y_{1}$), ($x_{2}$, $y_{2}$), ($x_{3}$, $y_{3}$), ($x_{4}$, $y_{4}$) such that: These coordinates satisfy the square geometric constraints (implemented through constraint conditions). And the sum of the weighted deviations from the various predicted coordinates is minimized.

3. Construct constraint conditions $C_{k}$(*x*) *k* = 1, *…*, 8

In our altitude prediction method, the ideal scenario is that the predicted coordinates from the cropped drone images of the top-left, bottom-left, top-right, and bottom-right corners are accurate and form a standard square for solving. Based on this condition, we construct geometric constraints for the predicted points.

The distances between predicted points ($x_{i}$, $y_{i}$) and the center point ($x_{c}$, $y_{c}$) are equal $C_{1}$(*x*) − $C_{4}$(*x*).

The distances between each pair of predicted points are equal $C_{5}$(*x*) − $C_{8}$(*x*).

4. Local SQP Iteration Steps

1. Initial point $X^{0}$, choose the average of the 12 predicted coordinates generated by each image after eliminating outliers. Set *k* = 0.

2. Update the Hesse matrix approximation *H*, with $H^{0}$ chosen as the identity matrix. The iteration method for $H^{k}$ uses the DFP method:(2)$$H^{k+1}=H^{k}+\frac{\Delta X^{k}{\left[\Delta X^{k}\right]}^{T}}{{\left[\Delta q^{k}\right]}^{T}\Delta X^{k}}-\frac{H^{k}\Delta q^{k}{\left[\Delta q^{k}\right]}^{T}H^{k}}{{\left[\Delta q^{k}\right]}^{T}H^{k}\Delta q^{k}}$$
where Δ*X* is the difference of each iteration of the target variable *X*, and Δ*q* is the difference in the gradient of each iteration.

3. Construct a QP sub-problem
(3)$$\begin{array}{c} \min\frac{1}{2}d^{T}Hd+g^{T}d\\ \;s.t.\;C_{i}\left(X^{k}\right)+\nabla C_{i}{\left(X^{k}\right)}^{T}d=0\;i=1,\ldots ,8 \end{array}$$
where *d* is the search direction, and *g* is the gradient of the objective function at the current point $X^{k}$.

4. Line search: Find step size α such that $X^{k+1}$ = $X^{k}$ + α × *d* minimizes the objective function value while satisfying the constraints.

5. Update: $X^{k+1}$ = $X^{k}$ + α × *d*

6. Check convergence condition: If the convergence condition is met (such as || $X^{k+1}$ − $X^{k}$ || < ε1 or |J($X^{k+1}$) − J($X^{k}$)| < ε2), stop; otherwise, return to step 2.

In each iteration, we solve a QP subproblem to find a search direction that both reduces the objective function value and satisfies the constraints.

## 4. Experimental Setup

### 4.1. Training Configuration

In this experiment, PCPVT-S was used as the backbone network for the MH-FPI model during the training process. PCPVT-S was pre-trained on the ImageNet dataset with its weights loaded, and the original classification layer was removed. Unless otherwise specified, the feature fusion network in MH-FPI employed the FPN network for fusion operations.

The AdamW optimizer was used to update the model parameters during training. The initial learning rate was set to $4\times 10^{-4}$, with a weight decay parameter of $5\times 10^{-4}$.

The model was trained for 45 epochs with a batch size of 16. To prevent overfitting, the learning rate was reduced by a factor of 10 at the 10th and 20th epochs.

The model was trained on the UL14plus dataset and validated on its test set. During training, the input image size for the UAV image branch was set to 256 × 256, while the satellite image map input branch used a size of 384 × 384. Data augmentation with random scaling and center point, as described in Section 2.3, was applied during training. All experiments in this section were conducted on an NVIDIA GeForce RTX 3060 GPU with mixed precision enabled.

### 4.2. Evaluation Metrics

Similar to OS-FPI, this study employed two evaluation metrics for the image-based geographic localization task: Relative Distance Score (RDS) and Meter-level Accuracy (MA) to assess the localization performance of the algorithm.

The Relative Distance Score (RDS) is an evaluation metric that assesses the relative distance between the algorithm’s predicted localization result and the ground truth on the satellite image. It is calculated by converting the relative distance between the predicted and true values into a score.

$d_{x}$ represents the pixel error in the horizontal direction between the predicted location and the true location, while $d_{y}$ represents the pixel error in the vertical direction between the predicted location and the true location. (Figure 10).
(4)$$\begin{array}{c} d_{x}=abs\left(X_{p}-X_{g}\right)\\ d_{y}=abs\left(Y_{p}-Y_{g}\right) \end{array}$$

After obtaining $d_{x}$ and $d_{y}$, the normalized pixel relative distance (*RD*) can be calculated. The value of *RD* ranges between 0 and 1.
(5)$$\begin{array}{c} RD=\sqrt{\frac{{\left(\frac{dx}{w}\right)}^{2}+{\left(\frac{dy}{h}\right)}^{2}}{2}} \end{array}$$

The Relative Distance Score (*RDS*) is derived by transforming the relative distance (*RD*) into a score between 0 and 1 using an exponential function. In this study, *k* is a scaling factor set to 10.
(6)$$RDS=e^{-k\times RD}$$

This study also employs Meter-level Accuracy (MA@K) as an evaluation metric for spatial distance. MA@K represents the proportion of samples in the entire dataset where the error between the algorithm’s predicted geographic location and the true geographic location is within K meters.

### 4.3. Experimental Results and Visualization of UAV Localization

We conducted experiments on our newly adopted multi-height dataset, UL14plus. The proposed MH-FPI model was compared with three backbone models: OS-FPI [6], WAMF-FPI [23], and FPI [6]. Figure 11 shows the accuracy rates for MA@3, MA@5, MA@10, MA@20, and MA@50 under the MA evaluation metric. MH-FPI demonstrated improved accuracy within 3 m, 5 m, 10 m, 20 m, and 50 m ranges. Notably, the 5 m and 10 m metrics showed improvements of 10.4% and 11.3%, respectively, compared to the OS-FPI model. Our model backbone, by comparing satellite images with two UAV images, achieved greater improvements in accuracy within the 5–20 m range. This result aligns with our hypothesis of first performing coarse localization using larger-scale UAV images, and then conducting fine localization using smaller-scale but more detailed UAV images within the area of interest. For images where the first coarse localization produces correct results, the second fine localization yields even more precise localization effects.

Table 1 presents the RDS scores, parameter counts, computational operations, and select meter-level accuracies for each model. Admittedly, compared to the OS-FPI model, MH-FPI shows an increase in both parameter count and computational load, primarily due to the two localization operations in the backbone. However, our proposed backbone is more suitable for UAV localization problems in multi-height scenarios. The comprehensive RDS score improved by 7.7 compared to OS-FPI. The experimental results indicate that our proposed multiple attention structure helps geographical localization network models learn height changes and position offsets between images, overcoming challenges in localization tasks and improving the accuracy of localization results.

Figure 12 displays the visualization of the MH-FPI network model’s geographical localization effect. The first row shows input UAV aerial images of the query location. The second row presents corresponding satellite image maps of the query area. The third row visualizes the feature heat maps output by the network model. The fourth row shows the geographical localization results predicted using the feature heat maps, with green circles indicating true geographical locations and blue circles representing the predicted geographical locations based on the model’s output feature heat maps. Observing these result images reveals that the network model has achieved high localization performance. In some images, it is challenging to provide accurate localization results quickly through visual observation alone, whereas our network model can provide relatively precise localization results.

### 4.4. Altitude Determination-Related Experimental Results

In our proposed fixed-height method, the core idea is to calculate the flight height by converting the area of the captured range, computed from ground coordinates, using a proportional relationship. Through Sequential Quadratic Programming (SQP), we comprehensively evaluate multiple predicted values generated from each coordinate point to calculate the optimal ground range.

In this study, each UAV aerial photograph is matched with 12 satellite images containing the UAV image at different spatial scales. Satellite images of different sizes produce varying degrees of accuracy during the matching process. These accuracies are reflected as weights in the objective function of the sequential quadratic programming, essentially seeking an optimal solution in space that both satisfies geometric constraints and fits all predicted data as closely as possible, embodying balance and compromise.

Figure 13 shows the partial meter-level (3 m, 5 m, 20 m) matching accuracies produced by satellite images of different sizes. The satellite image cropping range is from 700 to 1800, corresponding to real-world distances from 180 m to 463 m.

In the fixed-height calculation process, to comprehensively consider the coordinates produced by matching satellite images of different sizes and the varying accuracies of different matching results, Sequential Quadratic Programming (SQP) iteratively seeks the minimum value of the objective function J while satisfying geometric constraints. In the test set, the function converged after an average of 14 iterations. This is a gradual process of convergence.

After SQP optimization, while maintaining the accuracy of each predicted point, the irregular shape in the second image from the left in Figure 14 was dynamically adjusted to the ideal square shape we envisioned. Figure 15 below shows the relative errors of the original height calculation and the height after SQP processing.

Testing revealed that the fixed-height method achieves higher accuracy in the 140–160 m range. We believe this is because localization accuracy reaches an optimal level at this height, while the impact of image cropping on the model’s localization gradually decreases with increasing altitude. These two factors achieve a balance in the 140 m to 160 m interval.

The comprehensive relative height error obtained through direct calculation was 23.17%, which was reduced to 18.16% after SQP processing. This indicates that the SQP adjustment resulted in a shape formed by the predicted coordinate points that more closely approximates the real situation.

The formula for relative height calculation:(7)relativeerror=abs(Predictedheight−Trueheight)/Trueheight

## 5. Conclusions

UAVs are now widely used in military, civilian, and commercial fields, with autonomous positioning capability being core to their mission completion. Currently, UAVs mainly rely on global satellite positioning systems for geolocation. However, in actual flight operations, factors such as obstruction, electromagnetic interference, and adverse weather can affect the strength of satellite positioning signals received by UAVs. In extreme cases, this can lead to UAVs operating in GNSS-denied environments, where they lose autonomous positioning capability and cannot conduct flight operations.

Therefore, solving the positioning problem for UAVs in GNSS-denied environments has become a research hotspot. This paper uses artificial intelligence technology and deep learning visual algorithms to enable UAVs to obtain their current geographical location by matching ground images captured by their onboard cameras in GNSS-denied environments. The main research content and work summary of this paper are as follows:

(1) We proposed a multi-altitude, multi-style UAV image dataset, UL14plus, based on which we conducted training and testing for UAV image positioning and altitude determination.

(2) We proposed a vision-based positioning algorithm, MH-FPI, based on an attention mechanism. The network model design references related work in the field of object tracking. By inputting two UAV images with the same center point and a satellite image into the model, it uses the Twins-PCPVT-S backbone network for feature extraction. The satellite and UAV images undergo two rounds of matching, producing coarse and precise positioning. Subsequently, it outputs a feature heatmap, predicting the UAV’s position on the satellite image map based on the heat value distribution on the feature heatmap, thus obtaining the corresponding geographical coordinates. Ultimately, the MH-FPI network model achieved a positioning accuracy of nearly 81% within a 20 m range and an RDS score of 76.3 on the UL14plus dataset test set.

(3) We proposed a method for calculating UAV flight altitude through image analysis. By using the model to calculate four-point coordinates in a single UAV image, we derive the actual area of the UAV’s captured image and calculate the flight altitude based on the proportional relationship. Additionally, we introduced a sequential quadratic programming method to optimize the area calculation. The final relative error in altitude determination is 18.16%.

The positioning and altitude determination methods proposed in this paper have achieved good accuracy for multi-altitude scenarios. The method of directly estimating altitude from images is newly proposed, with its advantage being that each frame of UAV image can independently calculate altitude without relying on data support from other sensors. Currently, the accuracy achieved is not high, mainly due to positioning offsets leading to deviations in altitude calculations. The source of error is relatively singular, leaving room for future improvements. Ideally, further research is needed on the problem of enhancing altitude determination accuracy as a supplement and aid to UAV altitude sensors in GNSS-denied environments. Methods such as image optical flow and joint use of multiple frames of images could be considered.

## Figures and Tables

**Figure 1 sensors-24-05491-f001:**
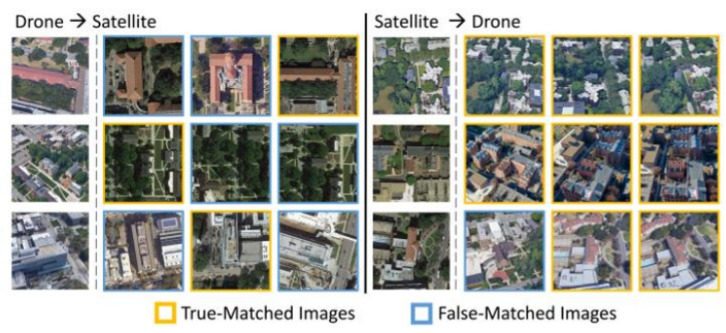
University 1652 dataset.

**Figure 2 sensors-24-05491-f002:**
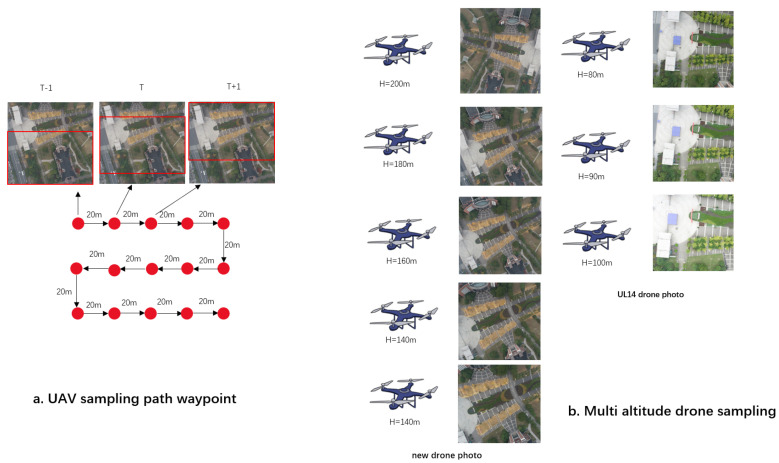
Sampling waypoints and altitude display.

**Figure 3 sensors-24-05491-f003:**
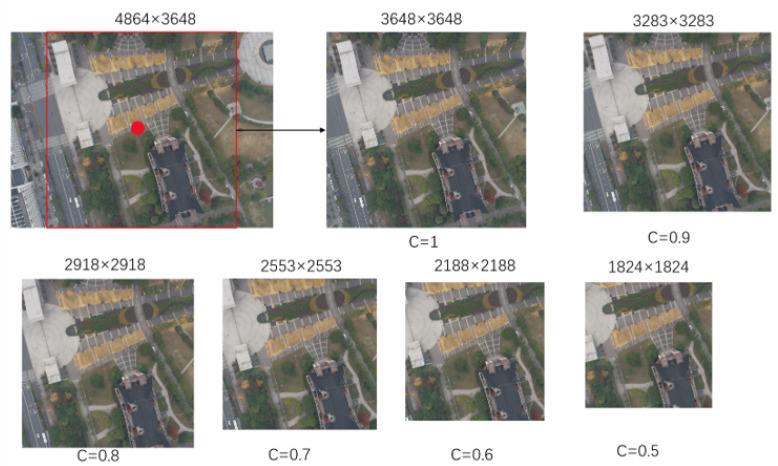
Cropped display of images collected at a height of 200 m.The red dot represents the center of the drone’s shooting, and the box represents the maximum square area that can be captured from the drone image.

**Figure 4 sensors-24-05491-f004:**
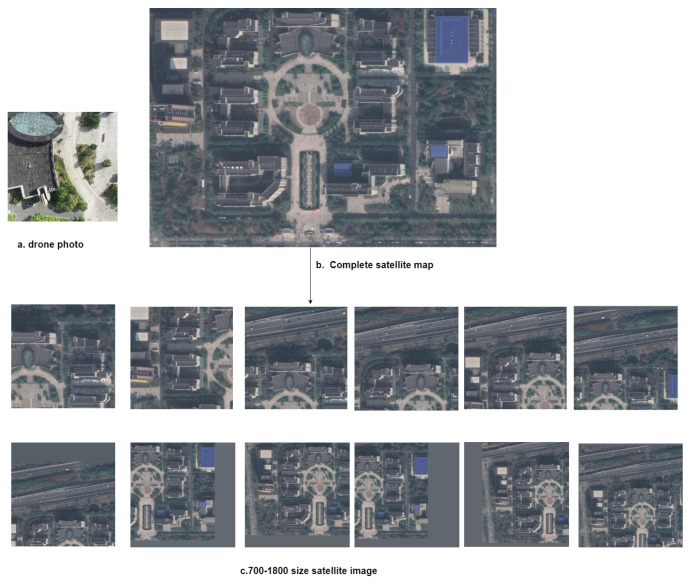
Satellite images of different sizes randomly shifted corresponding to the drone images in the test set.

**Figure 5 sensors-24-05491-f005:**
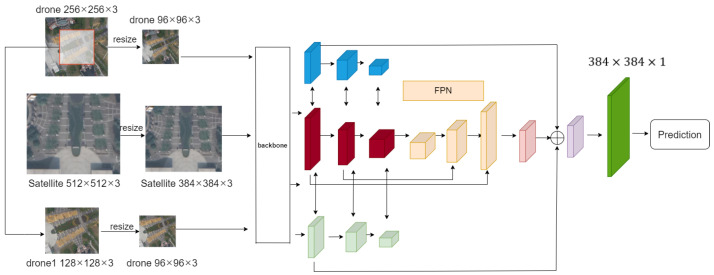
The backbone is MH-PCPVT, the neck is FPN, and the detection head restores the feature map to the size of the satellite image through linear interpolation.The blue part is the feature map produced by the large drone map. The red part is the feature map produced by the satellite image. The green part is the feature map produced by the cropped small drone map.

**Figure 6 sensors-24-05491-f006:**
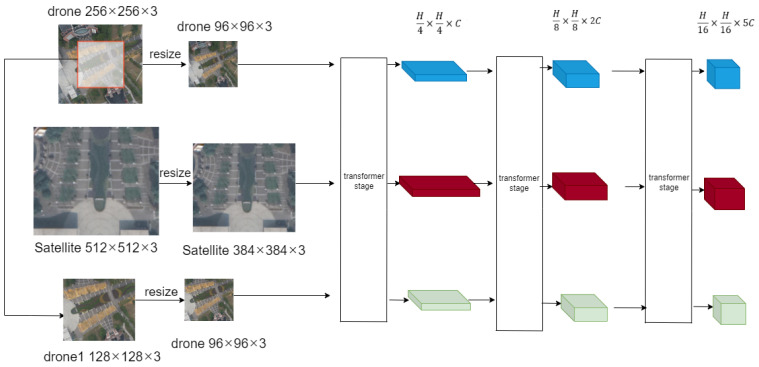
Feature extraction backbone. The features gradually decrease in size over the three stages, while the number of feature map channels for these three stages are 64, 128, and 320 respectively.

**Figure 7 sensors-24-05491-f007:**
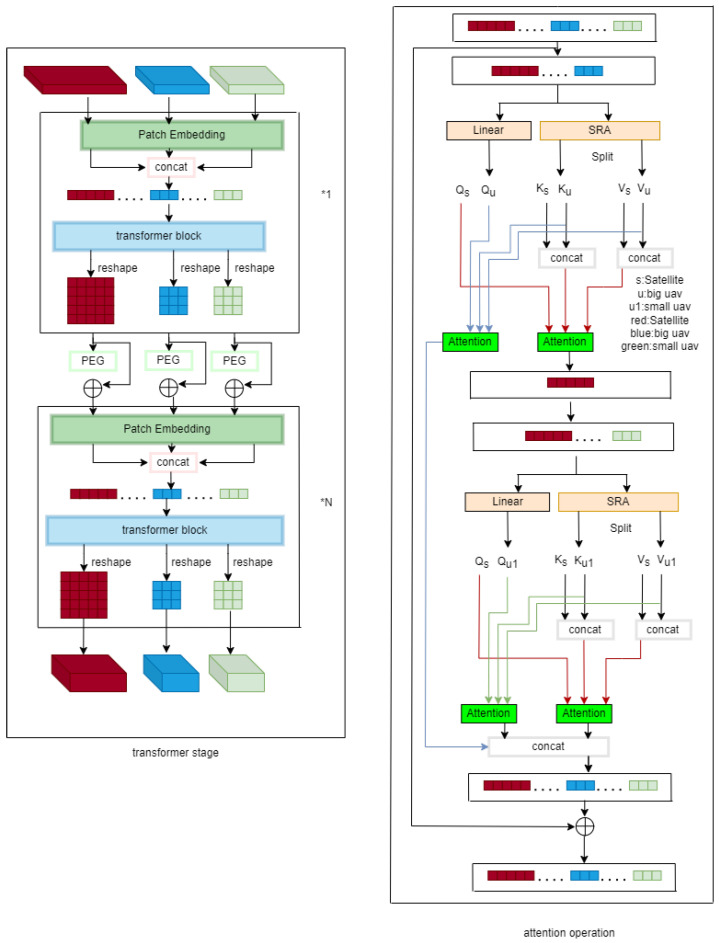
Transformer stage and attention operation.

**Figure 8 sensors-24-05491-f008:**
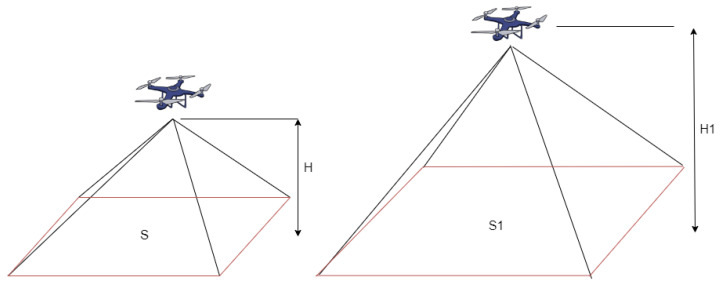
Drone shooting range and height. S: area, H: height.

**Figure 9 sensors-24-05491-f009:**
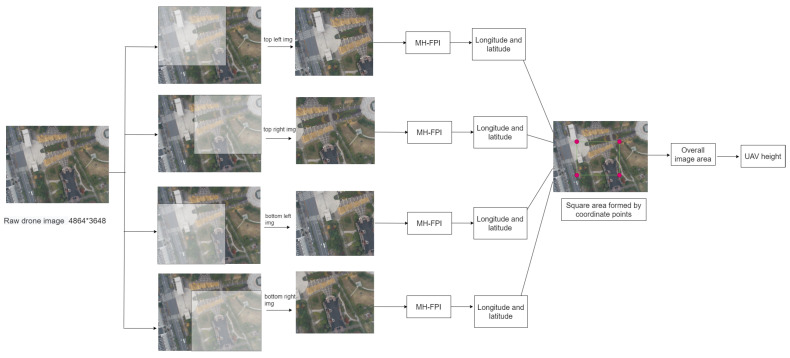
Height calculation process.Taking the four vertices of the original drone image as the starting point, four images of 3000 × 3000 size were intercepted.

**Figure 10 sensors-24-05491-f010:**
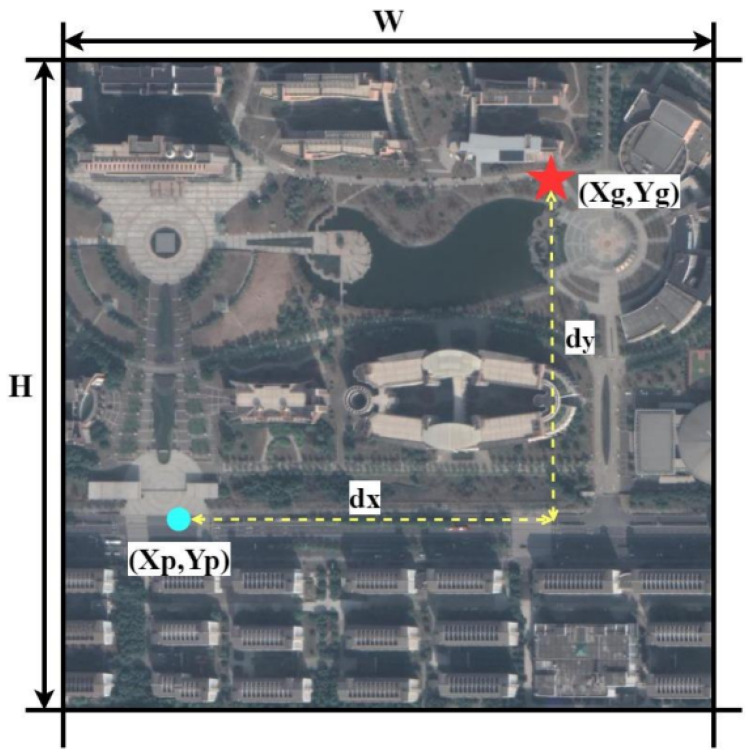
Relative distance map.

**Figure 11 sensors-24-05491-f011:**
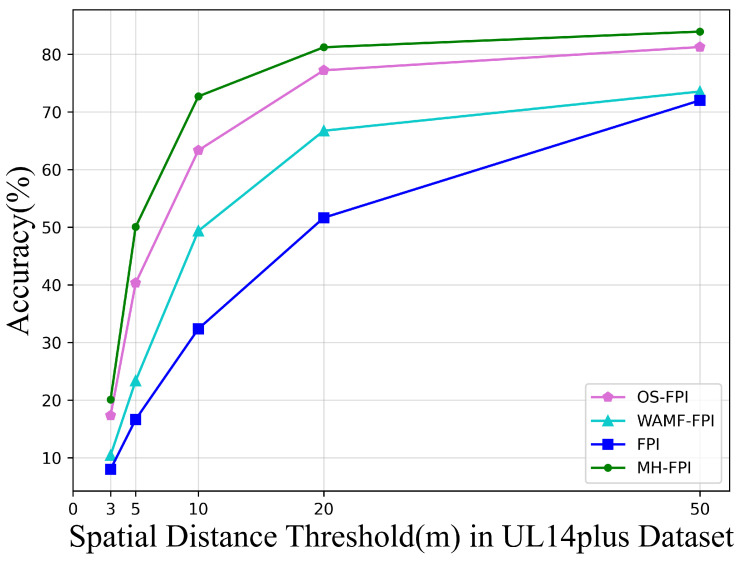
Meter-level accuracy of different model parts.

**Figure 12 sensors-24-05491-f012:**
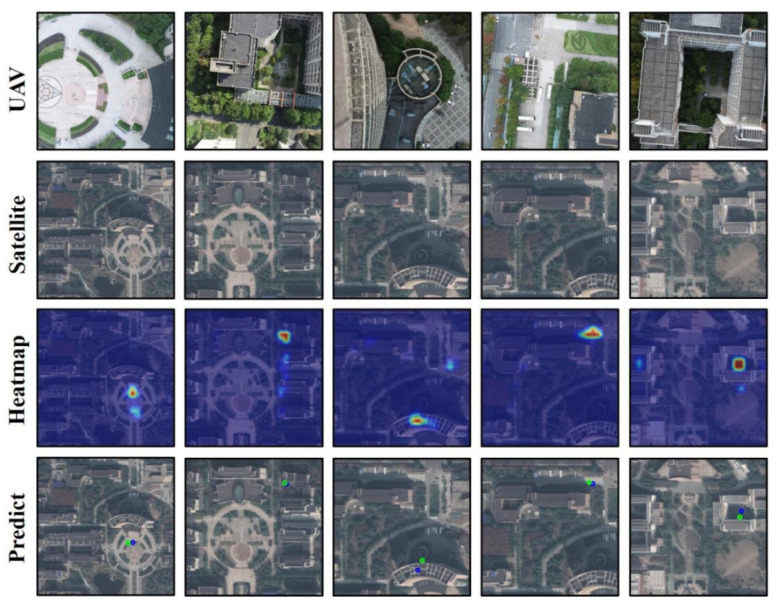
Visualization results.

**Figure 13 sensors-24-05491-f013:**
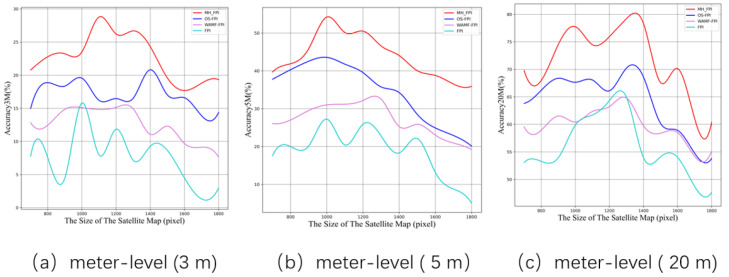
Meter-level matching accuracy produced by satellite images of different sizes.

**Figure 14 sensors-24-05491-f014:**
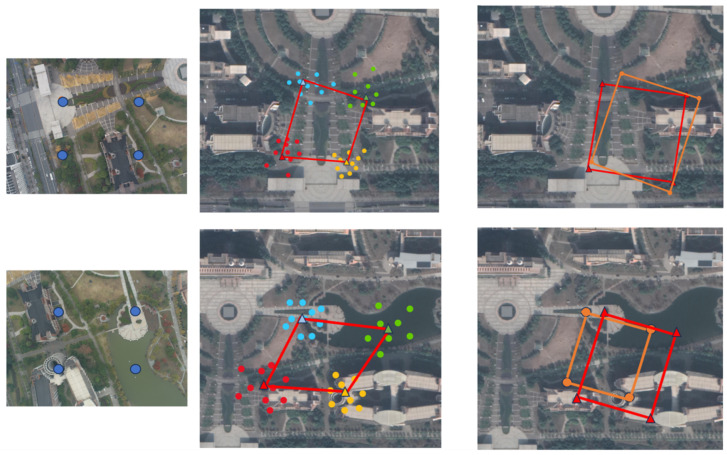
The first column is the original image of the drone, and the blue marked point is the center point of the drone mini image that is ready to be cropped. The second column shows the positions of multiple predicted coordinates generated by the cropped drone image on the satellite image (The four colored dots represent drone images that have been cropped into quarters, and each drone image predicts the resulting location, excluding outliers), the coordinates generated by taking the average, and the shape enclosed. The red part in the third column represents the predicted coordinates and the regular shapes enclosed by each drone small image with the minimum objective function after multiple predicted coordinates have passed through SQP. The orange part represents the true position of the center point coordinates of the drone image on the satellite map.

**Figure 15 sensors-24-05491-f015:**
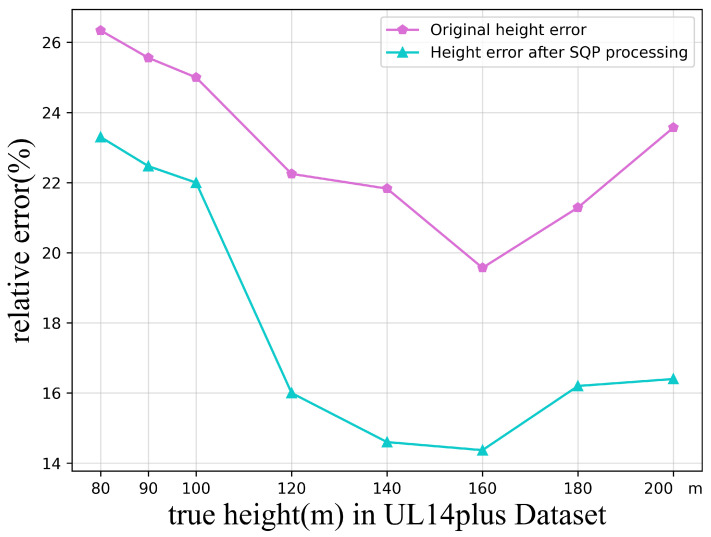
Relative error in calculating height at different altitudes.

**Table 1 sensors-24-05491-t001:** Different model parameters, calculation amount, RDS scores and partial meter-level accuracy.

Model	Parameter (M)	GFLOPS	RDS	<3 m (%)	<5 m (%)	<20 m (%)
MH-FPI	17.27	15.39	76.32	21.37	50.62	81.28
OS-FPI	14.76	10.42	68.57	17.34	40.19	76.52
WAMF-FPI	48.5	13.55	60.49	10.45	27.83	57.72
FPI	44.48	14.88	57.15	9.86	22.39	53.77

## Data Availability

No new data were created or analyzed in this study. Data sharing is not applicable to this article.

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
