# Peer review of "3D Positioning of Drones through Images"

_sensors, 2024, doi:10.3390/s24175491_

Round 1
Reviewer 1 Report
Comments and Suggestions for Authors
1. Acronyms need to be explained when they first appear in the paper, such as RDS, WAMF, OS-FPI and so on.
2.The format of the paper needs to be improved, for example, inconsistent line spacing and the fonts size in figures.
3.Each camera will have distortion at the edge of the captured image, which will lead to inaccurate real area calculation covered in the image. How to deal with this point should bet explained in details in this paper.
4.The experiments in this paper all scale the area after the test, that is, flying in the modeled area. What will happen if the algorithm runs in the reak unknown denial area with the satellite image that is not updated in time?
Comments on the Quality of English LanguageModerate editing of English language required.
Author Response
Comments 1:[Acronyms need to be explained when they first appear in the paper, such as RDS, WAMF, OS-FPI and so on.]
Response 1:[Thank you for pointing this out. We agree with this comment. In the first section of the introduction, we added an introduction to professional terminology and highlighted it in red]
Comments 2:[The format of the paper needs to be improved, for example, inconsistent line spacing and the fonts size in figures.]
Response 2:[Thank you for pointing this out. We will further communicate with the editor to modify the format to ensure the integrity of the article]
Comments 3:[Each camera will have distortion at the edge of the captured image, which will lead to inaccurate real area calculation covered in the image. How to deal with this point should bet explained in details in this paper.]
Response 3:[Thank you for your feedback. This has already been taken into consideration in our design. We tested the camera of the DJI Elf 4 drone used and concluded that the camera corrected distortion with a distortion rate of less than 1%. We verified the distortion of camera captured images by taking high-altitude shots of a field with equidistant grid lines. On the site, the fixed width of each grid is 4m, and drones take photos from three directions to verify the distortion in the horizontal, vertical, and inclined directions. Measure the pixels occupied by the grid on both sides of the photo and the grid in the center area of the photo through manual calibration. After calculation, the distortion is less than 1%. (The image obtained from the test is 4.8.png in the folder.)We believe that this level of distortion has a relatively small negative impact on the cropped image being fed into the neural network for image matching.Do we need to add this verification process to the article? I hope you can give us some advice]
Comments 4:[The experiments in this paper all scale the area after the test, that is, flying in the modeled area. What will happen if the algorithm runs in the reak unknown denial area with the satellite image that is not updated in time?]
Response 4:[Thank you for bringing this up. In our project, the model is first trained using pre captured area images, and then the model can be applied to unknown areas. Indeed, different weather conditions, ground environments, architectural styles, etc. can all affect the positioning accuracy of the model. So if the style of the unknown area differs too much from the image style provided when we trained the model, the accuracy of localization will decrease. The pursuit of a unified, multi scenario applicable model is our ongoing goal.]
Reviewer 2 Report
Comments and Suggestions for Authors 1) The paper can benefit from a better description and justification of different stages of the algorithm i.e. not only stating what is done by the reason behind it. 2) A section on the features that are extracted from the images is important and I suggest it be added to benefit readers. 3) Since the results are based on software, a section that describes how the software was implemented is needed. What language? What platform? what toolboxes.4)When there are multiple plots in figures. Label them as (a) (b)..etc and describe each plot.
5)Describe the features extracted in Fig. 5
Comments on the Quality of English Language
The manuscript requires editing.
Author Response
Comments 1: [ The paper can benefit from a better description and justification of different stages of the algorithm i.e. not only stating what is done by the reason behind it.]
Response 1: [Thank you for your suggestion. We made appropriate modifications and highlighted them in red in the third section, and added a Feature Extraction Network section in Section 3.]
Comments 2: [ A section on the features that are extracted from the images is important and I suggest it be added to benefit readers.]
Response 2: [I agree with your viewpoint. Therefore, we added the Feature Extraction Network in the third section. Thank you for your valuable feedback.]
Comments 3: [Since the results are based on software, a section that describes how the software was implemented is needed. What language? What platform? what toolboxes.]
Response 3: [Thank you for your suggestion. The question you raised has a long-term perspective, and the application of future projects will involve specific software development. However, the current project is still in the laboratory stage. All training and testing processes are implemented on PyCharm. The implementation details are discussed in section 4.1.]
Comments 4: [When there are multiple plots in figures. Label them as (a) (b)..etc and describe each plot.]
Response 4: [Thank you for providing constructive feedback on this point. We will make modifications to images 2.2, 2.4,4.4 and 4.6.]
Comments 5: [Describe the features extracted in Fig. 5]
Response 5: [Thank you for your suggestion. We have moved Figure 5 to the position of Figure 6 and placed it in the Feature Extraction Network section, which will been explained in this section.]
Round 2
Reviewer 1 Report
Comments and Suggestions for Authors
Thanks to the authors for their meticulous revision work.
Comments on the Quality of English LanguageMinor editing of English language required, such description of figures, algorithms and formulas.